# The Agreement between Patients’ and Healthcare Professionals’ Assessment of Patients’ Health Literacy—A Systematic Review

**DOI:** 10.3390/ijerph17072372

**Published:** 2020-03-31

**Authors:** Mona Voigt-Barbarowicz, Anna Levke Brütt

**Affiliations:** Department of Health Services Research, Faculty of Medicine and Health Sciences, Carl von Ossietzky University Oldenburg, 26129 Oldenburg, Germany

**Keywords:** HL, HCP, estimation, HL needs, physicians, patient-provider relationship, measurement

## Abstract

Healthcare professionals (HCPs) can play a key role in promoting health literacy (HL) in patients to help them navigate the healthcare system effectively. This involves assisting patients to locate, comprehend and evaluate health information. HCPs should assess patients’ health literacy needs and check the patient´s understanding to communicate adequate health information. This review investigates the agreement between the patients’ and HCPs assessment of patients’ HL. A systematic literature search in PubMed, Scopus, PsycINFO, CINAHL and the Cochrane Library was performed in November 2019. The search yielded 6762 citations, seven studies met the inclusion criteria. The following HL measurement instruments were completed by the patients in the included studies: REALM (*n* = 2), REALM-R (*n* = 1), S-TOFHLA (*n* = 1), NVS (*n* = 1), SILS (*n* = 1), HLSI-SF (*n* = 1) and HLS-EU-Q16 (*n* = 1). The HCPs assessed patients’ HL by answering questions that reflect the content of standardized tools. Six studies reported that a high proportion of patients assigned to have HL needs based on their self-report were overestimated by their HCPs in terms of the HL level. The results demonstrated that HCPs had difficulty determining patients’ HL adequately. Differences between the HL estimation of HCPs and the actual HL skills of patients might lead to communication problems.

## 1. Introduction

### 1.1. Health Literacy

Modern healthcare systems are becoming more complex, whereby patients are required to meet high demands in order to be able to adequately use health-relevant information. The health literacy (HL) of a patient depends not only on individual abilities and skills but also on the environment (i.e., the healthcare setting), which can impact the communication demands placed on patients. In particular, patients with HL needs have a major problem in navigating through today’s healthcare systems [1]. HL is defined as ‘…the cognitive and social skills that determine the motivation and ability of individuals to gain access to, understand and use information in ways that promote and maintain good health’ [2]. According to Nutbeam [3], HL can be subdivided into three levels that build on each other. The functional level includes basic reading and writing skills and being able to apply to health-related materials or verbal health instructions. A patient who reaches this level is able to absorb and understand simple health-relevant information. At the second level, the interactive level, the patient becomes a further active part of the healthcare system; extensive HL as well as cognitive and social skills are required. At the third level, the critical level, health information and existing recommendations are no longer arbitrarily accepted but increasingly questioned. At this level patients are already actively exchanging information with the existing healthcare system and are thus abandoning their passive role. 

HL is central to successful disease management and a prerequisite for the promotion of health-preserving and disease-preventing behaviour, for adequate handling of illnesses and for participation in the process of decision-making [4]. A growing number of studies link low patients’ HL with higher healthcare costs and poorer health outcomes, including but not limited to higher rates of hospitalization and poorer medication adherence, poorer physical and psychological health and poorer self-management of chronic diseases (e.g., hypertension, diabetes) [5,6,7,8,9]. Risk groups for having HL needs are non-native speakers, the elderly and those with limited education [10,11].

Patients have difficulties understanding technical terms and jargon, which are often embedded in medical communication during medical encounters [12]. Misunderstandings could arise when HCPs fail to identify the HL needs of patients. Overestimation of the patients’ HL by HCPs carries the risk that the patients will have difficulties understanding the information that the HCPs give when HCPs do not adapt their wording, e.g., medical terms, to the patients’ HL abilities [13,14,15]. Training for HCPs could broaden the understanding of HL and thus support the identification of patients´ HL needs by HCPs [13].

Being able to correctly assess the patient’s HL and to identify patients’ HL needs is a prerequisite for patient-centred HCP-patient communication. Strategies including improved communication skills and using teach back methods further promote successful HCP-patient communication [16]. Further information could be provided by using comprehensive HL measurement instruments as well as the different qualifications regarding their professions of the HCPs [17].

### 1.2. Measurements of Health Literacy

Measurement instruments assessing and identifying the HL skills, needs, strengths and preferences of patients are needed [18,19]. In recent years, standardized instruments have been developed to measure HL. Measurements can be either a direct test of an individual’s abilities and assess the actual performance in given tasks, e.g., solving tasks dealing with numeracy or print/oral literacy, or the assessment of self-reported abilities [19]. Frequently used examples of objective measurement instruments are the Rapid Estimate of Adult Literacy in Medicine (REALM) [20] and the revised version, REALM-R [21]. The REALM instrument is a reading recognition test composed of 66 health-related words and provides an estimate of an individual´s reading ability. The REALM demonstrated an internal consistency of *α* = 0.94 [22] and the REALM-R demonstrated a Cronbach’s *α* of 0.91 [17]. More comprehensive objective measurements are provided by the Health Literacy Skills Instrument (HLSI) [23] and the short form (HLSI-SF) [24], which combine an objective measurement and a self-report of HL abilities. The HLSI-SF [24] is a 10-item measurement that assesses the following domains of HL: (1) comprehension, (2) print literacy skills, (3) numeracy skills, (4) oral literacy skills and (5) information seeking skills [24]. The HLSI-SF demonstrated acceptable internal consistency reliability (*α* = 0.70), correlation with HL domains and S-TOFHLA and HSLI was of 0.47 (Sensitivity = 0.71; Specificity = 0.65) [24]. In subjective measurements, individuals self-report their own experiences and abilities. The self-report approach is applied in the Health Literacy Questionnaire (HLQ) [25]. The HLQ consists of 44 items, comprising nine separate scales to measure the multidimensional concept of HL. In most settings the HLQ demonstrated Cronbach’s *α* of > 0.8 for most scales [25].

In their review, Altin et al. [19] examine recently published (from 2009 forward) papers dealing with the development and evaluation of HL instruments. Newly developed instruments apply multidimensional measurement of HL, whereas instruments developed before 2009 focus on basic skills regarding functional HL. Functional HL measurements overlooked the complexity of a patients´ HL and did not consider contextual factors, personal values, social resources and individual motivations that influence a patients´ ability to understand and act upon health information [26,27].

In addition to self-reported HL, HCPs can also assess patient HL. The resident questions that were utilized in the study of Bass et al. [28] were designed to assess residents’ perception of patient HL levels. In this context the term ’residents’ refers to resident medical officers or physicians. These can assess whether their patients have a HL problem, what level the patients´ HL is and whether and how the patients´ HL will impact the visits.

The purpose of this systematic review was to gain an overview of HL assessment used in studies describing the agreement patients’ and healthcare professionals’ assessment of patients’ HL. Specifically, the review aimed to describe: (1) the assessment by patients and HCPs and (2) the agreement between patients’ and HCPs’ HL assessment.

## 2. Materials and Methods

We conducted a systematic review guided by the Preferred Reporting Items for Systematic Reviews to ensure transparent and complete reporting (Appendix A: PRISMA 2009 Checklist). The protocol was not registered.

### 2.1. Search Strategy

A systematic search for relevant studies in five databases (PubMed, Scopus, PsycINFO, CINAHL and Cochrane Library) was performed in November 2019. The searches were not limited to any time period or language. The keywords used in the search were health literacy, literate, nurse, physician, doctor, practitioner, health professional, HCP, therapist, clinician and patient. The search strategy was adapted for the different databases by applying the respective operators (Appendix B: Detailed search strategy).

### 2.2. Study Selection

Study selection was performed in two steps. Eligible studies were identified by title and abstract as well as full-text screening by two independent reviewers (M.M., M.V.B). In case of discrepancies, a third reviewer (A.L.B.) was consulted. At title and abstract screening (screening phase), studies were excluded according to the criteria listed in Table 1. At full-text screening (eligibility stage) further inclusion criteria were added (see Table 1). 

### 2.3. Study Extraction and Analysis

First, one reviewer (M.V.B.) extracted data from the included studies into a summary table. Afterwards, a second reviewer (A.L.B.) checked the entered data for accuracy and completeness. The summary tables included data on author, year, study design, objective, setting, number of patients, characteristics of HCPs, HL instruments used by the patients, questions/HL instruments used by the HCPs, categories of HL determined by HCPs, categories of HL determined by patients, HL assessment by patients, HL assessment by HCPs and results of agreement analysis. Subsequently, the assessment of patients´ HL by patients and HCPs was described. Finally, the agreement of and correlation between the HL assessment between patients and HCPs were described.

## 3. Results

### 3.1. Search Strategy and Studies

The search yielded 6762 citations; after removal of the duplicates, 3793 articles remained. Titles and abstracts were screened using predefined criteria (see Table 1, screening stage) for relevance, resulting in 42 publications that underwent full-text screening with regard to additional criteria (see Table 1, eligibility stage).

Of these, seven studies met the inclusion criteria and were included in this review. The reasons for exclusion at the eligibility stage were as follows: Missing assessment of patients´ HL by patients (*n* = 13) or HCPs (*n* = 11), articles did not report original research (*n* = 4), the measurement instrument used was not named (*n* = 4), the methodical approach was not reproducibly documented (*n* = 2) and patients and HCPs assessed not the same patient sample (*n* = 1), (Figure 1: PRISMA flow diagram). 

### 3.2. Study Characteristics

Details of the study characteristics are presented in Table 2. The included studies were published between 2002 and 2019. All studies were conducted in the USA (*n* = 6), except one study (*n* = 1) that originated from Belgium. Of the seven studies, six were cross-sectional studies; one study [29] had a prospective design. The studies took place in primary care (*n* = 4) or hospital-based care (*n* = 3) settings. The sample sizes ranged from 65 to 1375 patients and from 12 to 80 HCPs. The HCPs were physicians and nurses. The physicians worked in different settings and were differently specialized. 

### 3.3. Assessment of Patients’ HL by Patients and HCPs

The studies used different methods for the assessment of patients´ HL by patients and HCPs (Table 3). Patients completed the following HL measurement instruments in the included studies: Rapid Estimate of Adult Literacy in Medicine (REALM) [22] *n* = 2, Rapid Estimate of Adult Literacy in Medicine—Revised (REALM-R) [21] *n* = 1, short form of the Test of Functional Health Literacy in Adults (S-TOFHLA) [34] *n* = 1, Newest Vital Sign (NVS) [35] *n* = 1, Single Item Literacy Screener (SILS) [36] *n* = 1, Health Literacy Skills Instrument-Short Form (HLSI-SF) [24] *n* = 1 and the European Health Literacy Survey (HLS-EU-Q16) [37] *n* = 1. HCPs assessed patients’ HL based on single questions reflecting the content of standardized tools. Answer options were HL categories comparable to categories that could be derived from the patients’ measurement instrument. Due to the small number of included studies, the measurement instruments used by patients and the questions answered by HCPs for patients´ HL assessment are described per study.

In the study by Bass et al. [28], the Rapid Estimate of Adult Literacy in Medicine—Revised (REALM-R) [21] was used. The HCPs rated patients’ HL by reporting whether they feel a patient has a literacy problem or not [28].

The Newest Vital Sign (NVS) [35] and Single Item Literacy Screener (SILS) [36] instruments were used in the study by Dickens et al. [13] to assess patients’ HL by patients. After the patients completed the NVS, the HCPs were queried to estimate patients´ HL categories. For this, they selected one of three NVS questions that reflected the three HL categories of the patients. The three categories were high likelihood of limited literacy, possibility of limited literacy, and almost always adequate literacy [13].

Kelly and Haidet [30] and Lindau et al. [29] used the Rapid Estimate of Adult Literacy in Medicine (REALM) [20] to assess the patients´ HL by patients. The authors used different measurement instruments to assess the estimation of patients´ HL by HCPs. In the study by Kelly and Haidet. [30], the physicians assessed the patients’ HL categories on a scale corresponding to the REALM grade-equivalent HL levels. Lindau et al. [29] used a brief and anonymous questionnaire to ascertain physician perceptions of patients’ HL categories. The health professionals estimated the HL of their patients by answering the question, ‘‘Based on your interaction today, what is your estimate of your patient’s reading level?’’. Based on the raw score, the response included four HL categories: high school, 7th to 8th grade, 4th to 6th sixth grade, 3rd grade or below. For further analysis, the answer categories of patients and HCPs were dichotomized as adequate or inadequate [20].

The Short form of the Test of Functional Health Literacy in Adults (S-TOFHLA) [34] was used in the study by Rogers et al. [31]. A single question was used by health professionals to assess their patients´ HL. Their answers were rated with a 5-point Likert-type scale ranging from 1 (very poor understanding) to 5 (superior understanding).

Storms et al. [32] used the European Health Literacy Survey (HLS-EU-Q16) [37] to assess patients´ HL by patients. HCP estimation of their patients’ HL was restricted to a simple scale; HCPs choose between inadequate, problematic or adequate HL. To this end, HCPs were educated on the HL concept, HL categories and HLS-EU questionnaires. After that, the HCPs evaluated their patients’ HL according to the three categories.

The Health Literacy Skills Instrument-Short Form (HLSI-SF) [24] was used for patients´ assessment of patient HL in the study by Zawilinski et al. [33]. For the estimation of the patients´ HL, the HCPs chose a question that demonstrated patients´ HL categories (Table 3).

### 3.4. Assessment by Patients and HCPs and The Agreement between Patients’ and HCPs’ HL Assessment 

The results of patients’ HL as assessed by patients and HCPs, as well as their agreement, are presented in Table 3. Overestimation meant that the HCPs assessed the patients’ HL to be better than how the patients rated their own HL. In the case of underestimation, the HCPs assessed the patients’ HL to be lower than how the patients rated their own HL. In the study by Bass et al. [28], the REALM identified 41% of patients as having HL problems, whereas 90% of patients had adequate HL according to the HCPs perception. In the analysis of the assessment, the HCPs overestimated HL in more than a third (36%) of the patients. The continuity-adjusted chi-square [(1 df) = 13.18, *p* < 0.001] demonstrated a statistically significant overestimation of patients’ HL by HCPs.

In the study of Dickens et al. [13], the ratings of patients resulted in more than half (63%) with high likelihood of having limited HL. On the basis of the NVS scores, adequate HL was reported by 22% of the patients. HCPs assigned adequate HL to 22% of patients (overestimation), although self-report resulted in low HL. The kappa statistic (κ = 0.09) showed a very low level of agreement between patients´ NVS scores and the estimated patient HL by the HCPs.

Kelly and Haidet [30] assessed the patients´ on the basis of four categories, with category 4 indicating the highest HL. Compared to the patients´ assessments according to REALM scores, the HCPs overestimated one-fourth (25%) of the patients in total. A slightly smaller proportion of patients (15%) were underestimated by HCPs in terms of their HL. The exact agreement overall was 0.61. The kappa statistic (κ = 0.19, *p* < 0.01) showed a poor level of agreement between patients´ self-reported REALM categories and the patient HL assessment by HCPs.

Of the patients studied by Lindau et al. (2006) [29], the REALM score indicated that 35% had inadequate HL, while the HCPs assessed that 41% of the patients had inadequate HL. The kappa statistic (κ = 0.43, *p* = 0.0006) showed a high level of agreement between patients´ and HCPs estimations.

In the study by Rogers et al. [31], more than 75% of patients assessed their own HL as adequate. The assessment of the HCPs showed that 30% of patients had inadequate/marginal HL. Compared to the S-TOFHLA scores, HCPs overestimated more than half of patients with inadequate HL and underestimated a quarter of patients with adequate HL.

Storms et al. [32] used the HLS-EU-Q16 to assess patients´ HL. According to the scores achieved, more than 60% of patients had adequate HL, whereas the HCPs assigned 90% of their patients to adequate HL categories. Consequently, one-third (34%) of patients with problematic/inadequate HL were overestimated. In six out of ten patients, HCPs assigned the same HL category that the HLS-EU self-report resulted in. Based on these results, there was a slight agreement between the assessment of patients´ HL by patients and HCPs (κ = 0.033, 95% CI, 0.00124 to 0.0648, *p* < 0.05).

In the study by Zawilinski et al. [33], the HLSI-SF results indicated that 45% of the total sample had HL needs. HCPs assessed that more than a third (34%) of the patients had inadequate HL. Due to the high proportion of overestimations of patients´ HL by HCPs, the kappa statistic indicated a poor agreement between the patients´ and HCPs assessment of patients’ HL.

### 3.5. Factors Associated with (Dis)Agreement in Health Literacy Assessment

In the study by Storms et al. [32], patients´ HL is more likely underestimated by HCPs for patients who consulted with their general practitioner less than 1 year, between 1 and 5 years, between 6 and 10 years than for patients who consulted with their general practitioner for more than 10 years. HCPs underestimated the HL of patients with whom they had a relatively short HCP-patient relationship (less than 1 year) compared to the patients with whom they had a professional relationship for more than 10 years. Lindau et al. [29] described a significant association of physician predictions with patient follow-up adherence. The (dis)agreement between patient and HCP estimations differed significantly by the patients’ educational attainment [13,31,32]. The odds of underestimating and overestimating HL were higher for patients who had no education, primary education or secondary education compared with patients with higher education [32]. Male HCPs underestimated patients’ HL more often than female HCPs [32].

## 4. Discussion

The aim of this systematic review was to describe and summarize the agreement of patients´ HL as assessed by patients and HCPs. Studies in which the patients themselves and the HCPs assessed the HL of the same patient sample were reviewed.

In six out of seven included studies, the assessment of patients’ HL by patients and HCPs varied substantially. These six studies reported that the level of HL was overestimated by HCPs in a high proportion of patients having HL needs based on their self-reports (22–58% of patients). In addition, an underestimation of patients with adequate HL by HCPs was described, but with a lower percentage (5–29% of patients). Most of the reviewed studies either concluded a significant overestimation by HCPs or a poor agreement between patients’ HL assessment by patients and HCPs. The agreement of the estimation of patients’ HL by patients and HCPs was determined differently.

The different patient samples investigated in the included studies used hospital-based or primary care populations. The patients’ diseases were all somatic. Studies of patients with mental illnesses were not included in this review. HCPs were mostly represented by physicians (in six of seven studies).

On-the-job training for physicians who have difficulties identifying the HL of their patients is necessary [17]. This training should not only improve the identification of patients´ HL needs. There are further strategies to promote communication between HCPs and patients. These include the promotion of communications skills in general but also specific techniques, such as teach back. By the teach back method [38] HCPs check patients’ understanding by asking them to state in their own words what has been explained to them by the HCP. Using this method, the patients’ understanding can be confirmed [39].

HL training for HCPs increased their knowledge and awareness of patients´ HL-related problems [40,41]. Recently, as part of an EU project, communication trainings focused on HL and designed specifically for HCPs in hospital settings were developed and successfully tested in the Netherlands, Ireland and Italy [42,43]. The project showed that participation in these trainings subjectively improved HCP knowledge about HL, understanding of HL needs, awareness of their jargon; the training also improved self-efficacy and resulted in adaptations in patient interactions. Practice improvement and oral and written communication skills training have been guided by experiential findings [42,44]. Within the EU Improving PAtient Centered Communication Competences (IMPACCT) project, communication training is being further developed for use in the training of medical students [43]. HCPs may even benefit from short training programs. Ogrodnick et al. [45] piloted a one-hour HL and teach-back skills training session in a group of respiratory therapists; knowledge and communications scores increased significantly.

The studies included in this review mostly explored the agreement between physicians’ and patients’ estimation of patient HL. Future studies should investigate other HCP groups, such as nurses or physiotherapists. Due to different educational background and access to patients, assessment of patients´ HL needs in these HCP groups may vary. Moreover, hospital and primary care settings were focused. There were no studies available that took place in other settings such as rehabilitation where HCPs spend more time with the patient [46].

An important point to reflect on is the use of various standardized measurement instruments for the assessment of patients´ HL in the included studies. The instruments used measured different dimensions of HL [3]. Some measurement instruments covered the functional HL levels such as basic reading and writing skills (i.e., REALM [22], REALM- R [21], S-TOFHLA [34]), each providing evidence of psychometric properties. Other measurement instruments (i.e., HLSI-SF [24], HLS-EU-Q 16 [37], NVS [35]) that also have been psychometrically evaluated, covered aspects of the interactive level, such as cognitive and social skills, or the critical HL level, such as the ability to analyse health information. However, none of the measurement instruments completely covered all three HL levels. Over many years definitions for HL have evolved [3]. Measurement of HL has proved complex because HL consist of different domains, skills and abilities. The instruments used in the included studies were mostly one-dimensional and did not reflect the different domains of patients’ HL. The aim of using the instruments in the included studies was to assign a high or low HL to patients. To measure independent domains of HL, we need multidimensional instruments. A multidimensional instrument is the Health Literacy Questionnaire (HLQ) [25,47]. The HLQ comprises nine scales that each measure a domain of the multidimensional construct of HL [25]. Profiles can provide information about HL needs and strength. HCPs can use these profiles to better communicate and support the planning and realization of interventions. In a clinical context, HCPs can also use the Conversational Health Literacy Assessment Tool (CHAT) [48] to identify patients’ multidimensional HL needs or preferences. Based on the domains of the HLQ, the CHAT was designed to support HCPs to use ten open-ended questions (e.g., Who do you usually see to help you look after your health?) to have a structured conversation with patients that target five HL areas.

It should be noted that one-dimensional instruments could affect the assessment of patients´ HL by HCPs. The different levels of HL may also be differently assessible by HCPs in the medical encounter: The functional HL level skills (e.g., reading abilities sufficient to understand and realize health information) may not be aspects that HCPs can observe in and estimate based on the medical encounter. As patients communicate verbally, patients’ HL needs identified by instruments that focus on reading skills (i.e., REALM-R) but not aspects of verbal communication, may not be detected by HCPs. Furthermore, HCPs may have their own conceptions of HL. HCPs’ conceptions of HL could focus on specific domains of the multidimensional construct of HL. Which domains the HCPs’ assessment is based on, cannot be determined by asking questions such as ‘‘Does your patient have low health literacy?’’. Moreover, HCPs may assess HL based on a specific domain of HL (e.g., Feeling understood and supported by healthcare providers). This may not be covered by the HL instruments (i.e., REALM-R) which were completed by patients. The multidimensional construct of HL should also be reflected when HCPs identify HL needs and strengths of their patients. Needs can be identified in one scale, there might be HL resources in another scale.

Future studies may use more comprehensive HL measures, such as the Health Literacy Questionnaire (HLQ) [47]. The agreement between the assessment of patients’ HL by patients and HCPs may vary between the different scales. It should be tested whether the agreement between the assessment of patients’ HL by patients and HCPs is changed or even improved by the use of multidimensional instruments. This was investigated in a study by Hawkins et al. (2017) [47]; patients and clinicians completed the HLQ and were interviewed with regard to the reasons for their answers. The highest concordance was reported between the HLQ scales “Feeling understood and supported by healthcare providers” (80%) and “Actively managing my health” (69%). The highest discordance was seen in the HLQ scales “Ability to actively engage with healthcare providers” (56%) and “Social support for health” (49%). The scale “Understand health information well enough to know what to do”, an example of functional HL level assessment, had a concordance of 40% and a discordance of 37% between the assessment by patients and clinicians. Reasons for the discordance included different understanding of the items and assignment of scores, varying reliance on HCPs, and learning or development in patients. The HLQ may be used for detecting discordance between patients’ and HCPs’ estimations of patient HL, which may lead to better communication.

### Strengths and Limitations

The strength of the present review was the comprehensive search for studies specifically investigating the agreement of patients’ HL as assessed by patients and HCPs. A further strength was the structured selection and extraction procedure, which was conducted by three reviewers.

A limitation was that only pertinent literature databases were searched, and all included articles were published in peer reviewed journals. There may be further studies in the grey literature, which we have not taken into account in our review. Additionally, only studies that used quantitative approaches were included. This is a limitation because studies using qualitative methodology were not considered. As a consequence, information on the reasons behind patient and HCP estimations of patient HL were not a focus in this review.

Furthermore, data from the included studies were not extracted independently by the reviewers, as one reviewer (M.V.-B.) extracted the data and the second reviewer (A.L.B) checked them. Dis/agreement was not assessed. 

A further limitation of this review was the lack of quality assessment of the included studies. As we found no specific quality assessment instrument for the reviewed study types, and since the study number was already rather small, we decided not to assess study quality in this review. Overall, we conclude that the sample size and assessment quality was rather poor.

## 5. Conclusions

This systematic review summarizes the available literature and provides an overview on the topic of agreement between patients’ HL assess by patients and HCPs. The results reveal that HCPs could not identify HL problems in their patients and overestimated the patients´ HL. As a consequence, HCPs might communicate to the patients in such a manner that the patients do not understand the information. Communication trainings could help to improve communication between patients and HCPs. These trainings could provide a practical way to improve the assessment of the perceived patients’ ability to perform functional, interactive and critical HL tasks. Furthermore, trainings need to impart knowledge on communication skills including checking understanding, as well as ways to adequately identify HL needs to increase the HCPs’ capacity to identify and respond to patients’ HL needs.

## Figures and Tables

**Figure 1 ijerph-17-02372-f001:**
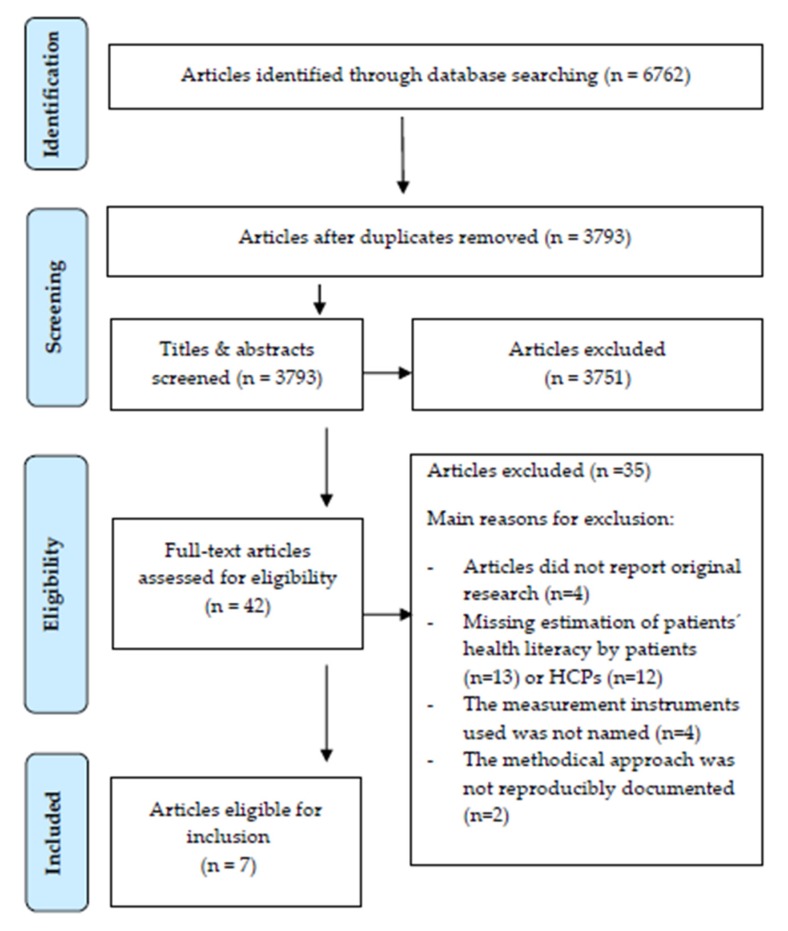
PRISMA flow diagram.

**Table 1 ijerph-17-02372-t001:** Exclusion and inclusion criteria.

Exclusion Criteria: Title and Abstract Screening (Screening Stage)	
EC 1	Participants were not at least 18 years old.
EC 2	Studies did not report original research.
EC 3	Studies did not assess health literacy.
EC 4	Studies did not use a quantitative approach for the estimation of HL.
EC 5	Patients were not involved in the assessment of HL.
EC 6	HCPs were not involved in the assessment of HL.
**Inclusion Criteria: Full-Text Screening (Eligibility Stage)**	
**Article Requirements**	
IC 1	Studies were published in a peer reviewed journal.
IC 2	Studies reported original research.
IC 3	The methodology was reproducibly documented.
**Assessment Requirements**	
IC 4	Studies addressed the analysis of agreement between patients’ HL assessment by patients and HCPs (i.e., providers that are in direct contact with patients).
IC 5	Patients (self- assessment) and HCPs (proxy assessment) assessed the same patient sample.
IC 6	HCP-rated patients’ HL measurement: HCPs assessed patients´ HL needs.
IC 7	Self-reported patients’ HL measurement: Studies used a standardized measurement of patients’ HL.

**Table 2 ijerph-17-02372-t002:** Study characteristics.

Author, Year, [Ref]	Area	Study Design	Aim/Objective	Setting	Number of Patients	Characteristics of HCPs
Bass et al., 2002, [28]	USA	Cross-sectional	To determine whether residents could identify patients with poor literacy skills based on clinical interactions during a continuity of care clinic visit.	Hospital-based care; (Internal medicine)	*n* = 182	Physicians *n* = 45
Dickens et al., 2013, [13]	USA	Cross-sectional	To compare nurses’ estimate of a patient’s HL to the patient’s HL.	Hospital-based care (Cardiac units)	*n* = 65	Nurses *n* = 30
Kelly & Haidet, 2007, [30]	USA	Cross-sectional	To investigate physician overestimation of patient literacy level in a primary care setting.	Primary care, Veterans Affairs Medical Center	*n* = 100	Physicians *n* = 12
Lindau et al., 2006, [29]	USA	Prospective	To examine the hypothesis that literacy predicts patient adherence to follow-up recommendations after an abnormal papsmear.	Primary care; Medical Center (HIV Obstetrics/ Gynaecology)	*n* = 68	Physicians *n* = 32
Rogers et al., 2006, [31]	USA	Cross-sectional	To determine whether primary care physicians can accurately identify patients who have limited understanding of medical information based solely on their clinical interactions with patients during an office visit.	Primary care (Family medicine)	*n* = 140	Physicians; *n* = 8 (second-year), *n* = 10 (third-year)
Storms et al., 2019, [32]	Bel-gium	Cross-sectional	To explore the agreement between patients’ HL and GPs’ HL estimations thereof, as well as to examine characteristics impacting this HL (dis)agreement.	Primary care (General practice)	*n* = 1469 (*n* = 1375 for analysis)	Physicians *n* = 80
Zawilinski et al., 2019, [33]	USA	Cross-sectional	To replicate and extend the findings of previous research by examining residents’ ability to predict HL levels in patients and to use a newer validated measure of HL.	Hospital-based care (Internal Medicine, Obstetrics/Gynaecology)	*n* = 38	Physicians n = 20

**Table 3 ijerph-17-02372-t003:** Assessment and agreement of patients’ HL as evaluated by patients and HCPs.

Author, Year [Ref]	HL Instrument–Patients	Questions/HL Instrument and HL Categories–HCPs	HL Categories–Patients	HL Assessment by Patients	HL Estimation by HCPs	Results
Bass et al. 2002 [28]	REALM-R^a^	“Do you feel this patient has a literacy problem?”; HCPs answered with “yes. (=Inadequate HL)” or “no. (=Adequate HL)”.	Inadequate HL: Scoring > −6; Adequate HL: Scoring > −7	*n* = 74 (36%); *n* = 108 (59%)	*n* = 18 (10%); *n* = 164 (90%)	Overestimation by HCPs: *n* = 59 (36%); Underestimation by HCPs: *n* = 3 (17%); Agreement: continuity-adjusted chi-square [(1 df) = 13.18, *p* < 0.001]
Dickens et al. 2013 [13]	NVS^b^, SILS^c^	NVS: High likelihood of limited literacy: “Does your patient have low health literacy?”; Possibility of limited literacy: “Does your patient have marginal health literacy?”; Almost always adequate literacy: “Does your patient have adequate health literacy?”	NVS: High likelihood of limited literacy, Possibility of limited literacy, Almost always adequate literacy; SILS: Inadequate HL: 1 (not at all), 2 (a little bit), 3 (somewhat); Adequate HL: 4 (quite a bit), 5 (extremely)	NVS: *n* = 41 (63%); *n* = 10 (15%); *n* = 14 (22%) SILS: *n* = 35% *n* = 65%	NVS: *n* = 19%, *n* = 13%, *n* = 68%	Overestimation by HCPs: *n* = 14 (22%); Agreement: kappa statistic, κ = 0.09
Kelly & Haidet 2007 [30]	REALM^d^	Scale corresponding to the Rapid Estimate of Adult Literacy in Medicine (REALM)	Level 1: 3rd grade and below; Level 2: 4th–6th grade: Level 3: 7th–8th grade; Level 4: High school	*n* = 4 (4%), *n* = 11 (11%), *n* = 47 (47%), *n* = 38 (38%)	Level 4: *n* = 74 (74%)	Overestimation by HCPs: *n* = 25 (25%), Underestimation by HCPs: *n*: *n* = 15 (15%); Agreement: kappa statistic, κ = 0.19 (*p* < 0.01), Level 1: 0.00, Level 2: 0.29, Level 3: 0.19, Level 4: 0.85, all levels: 0.61
Lindau et al. 2006 [29]	REALM^d^	Self-administeredquestionnaire: ‘‘Based on your interaction today, what is your estimate of your patient’s reading level?’’	Adequate (REALM ≥ 61 or high school level) or Inadequate (REALM ≤ 60 or below high school level)	*n* = 24 (35%), *n* = 44 (65%)	*n* = 23 (41%), *n* = 33 (59%)	Agreement: kappa statistic, κ = 0.43 (*p* = 0.0006)
Rogers et al. 2006 [31]	S-TOFHLA^e^	“What is your perception of the patient’s medical understanding?”, 5-point Likert-type scale ranging from 1 (very poor understanding) to 5 (superior understanding).	Inadequate HL: 0–16, Marginal HL: 17–22, Adequate HL: 23–36	*n* = 34 (24%), *n* = 106 (76%)	*n* = 42 (30%), *n* = 98 (70%)	Overestimation by HCPs: *n* = 18 (53%) Underestimation by HCPs: *n* = 26 (25%)
Storms et al., 2019 [32]	HLS-EU-Q 16^f^	HCPs were restricted to indicating that their patients’ HL was inadequate, problematic, or adequate HL on a simple scale.	4-point Likert scale (very difficult; difficult; easy; and very easy). These scores were dichotomized. Inadequate HL: 0–8, Problematic HL: 9–12, Adequate HL: 13–16	*n* = 201 (15%), *n* = 299 (22%), *n* = 875 (64%)	*n* = 1241 (90%), *n* = 130 (10%), *n* = 4 (<1%)	Overestimation by HCPs: *n* = 199 + 271/1375 (34%), Underestimation by HCPs: *n* = 68/1375 (5%); Agreement: kappa statistic, κ = 0.033, 95% CI, 0.00124 to 0.0648, *p* < 0.05; Correct estimation by HCPs: *n* = 837 (61%);
Zawilinski et al., 2019 [33]	HLSI-SF^g^	Resident Questionnaire (RQ): “Does the patient have a health literacy problem?”, “What is the patient’s level of health literacy?”,“Did patient’s health literacy impact the visit?”; Question 1+3: “yes,” “no,” or “not sure”, Question 2: “inadequate,” “adequate,” or “not sure”	Each correct response was given 1 point. Adequate HL: 7-10, Inadequate HL: 0-6	*n* = 21 (55%), *n* = 17 (45%)	*n* = 25 (66%), *n* = 13 (34%)	Overestimation by HCPs: *n* = 10 (58%), Underestimation by HCPs: *n* = 6 (29%); Agreement: kappa statistic, κ = 0.13, *p* = 0.42

^a^ Rapid Estimate of Adult Literacy in Medicine—Revised; ^b^ Newest Vital Sign; ^c^ Single Item Literacy Screener; ^d^ Rapid Estimate of Adult Literacy in Medicine; ^e^ Short form of the Test of Functional Health Literacy in Adults; ^f^ European Health Literacy Survey- Q16; ^g^ Health Literacy Skills Instrument-Short Form.

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
