# Peer review of "The Agreement between Patients’ and Healthcare Professionals’ Assessment of Patients’ Health Literacy—A Systematic Review"

_ijerph, 2020, doi:10.3390/ijerph17072372_

Round 1

Reviewer 1 Report

There is no measure of agreement between the two coders in the extraction stage. If no record of initial dis/agreement remained, pls refer to it as a limitation (in the discussion). The screening stage in the PRISMA needs to be tied (maybe in the text) to Table 1, so that the exclusion in the screening stage is made clear. Pls specify the N of all exclusion reasons. I would recommend considering the Agency for Healthcare Research and Quality (AHRQ) guidance (Owens et al 2010) for quality assessment.

Author Response

Response to Reviewer 1 Comments

  1. There is no measure of agreement between the two coders in the extraction stage. If no record of initial dis/agreement remained, pls refer to it as a limitation (in the discussion).

Dear reviewer, thank you for your comment. We took up your suggestion and revised the description of the extraction stage as follows:

“First, one reviewer (M.V.B.) extracted data from the included studies into a summary table. Afterwards, a second reviewer (A.L.B.) checked the entered data for accuracy and completeness. This was not an independent extraction of the two reviewers.”

In the discussion section, we added the following text:

“Furthermore, data from the included studies was not extracted independently by the reviewers, as one reviewer (M.V.B.) extracted the data and the second reviewer (A.L.B) checked them. Dis/agreement was not assessed.”

  1. The screening stage in the PRISMA needs to be tied (maybe in the text) to Table 1, so that the exclusion in the screening stage is made clear.

Thank you for the comment. We now refer to table 1 and the PRISMA flow diagram and revised the manuscript as follows:

“At title and abstract screening (screening phase), studies were excluded according to the criteria listed in Table 1. At full-text screening (eligibility stage) further inclusion criteria were added (see Table 1).”

  1. Pls specify the N of all exclusion reasons.

We specified the N of all exclusion reasons and corrected a mistake in the manuscript.

“Titles and abstracts were screened using predefined criteria (see Table 1, screening stage) for relevance, resulting in 42 publications that underwent full-text screening with regard to additional criteria (see Table 1, eligibility stage). Of these, seven studies met the inclusion criteria and were included in this review. The reasons for exclusion at the eligibility stage were as follows: Missing assessment of patients´ HL by patients (n=13) or HCPs (n=11), articles did not report original research (n=4), the measurement instrument used was not named (n=4), the methodical approach was not reproducibly documented (n=2) and patients and HCPs assessed not the same patient sample (n=1), (Figure 1: PRISMA 2009 flow diagram).”

  1. I would recommend considering the Agency for Healthcare Research and Quality (AHRQ) guidance (Owens et al 2010) for quality assessment.

Dear reviewer, thank you for recommendation of the AHRQ guidance. Unfortunately, not all items of this and other checklists really match our study sample. Therefore, quality assessment of the seven included studies would only partly add information. Due to the short time schedule with regard to the revision of the manuscript (10 days) we gave priority to the revision of other complex reviewer requests.

Reviewer 2 Report

Thank you for your review of the literature on this important topic. While previous research has already established that HCP overestimate the HL abilities of their patients, it is useful to have a current review of published studies. 

Abstract 

Line 13-14: Given the findings of your review and previous studies of the accuracy (or lack thereof) of which HCPs estimate the health literacy of patients, it is perhaps not reasonable to argue that HCPS should be able to adequately estimate (guess) the health literacy needs of their patients. It is more reasonable to argue that HCPs should ensure they adequately communicate health information and check patient understanding. 

Key words: Suggest that you review these to increase the likelihood of your article being found by those interested in this topic. Consider those that you used to search for articles to include in your review. Avoid overlap with words in the title of your article. 

Line 39-40: Check the explanation of functional health literacy. It is not limited to the ability to read and write - it also includes the ability to follow verbal instruction. 

Line 44: Check the final sentence of the paragraph to ensure that it links to critical health literacy.

Line 46: HL is central to (delete for)

Line 53-54: Consistent with contemporary understandings of health literacy, it would be more appropriate to refer to the purpose of measurement tools to identify the health literacy needs, strengths and preferences of patients rather than identify patients with low health literacy. 

Line 69-73: Please provide a reference for the RQ - the reference provided (17) does not mention the RQ. It may add to the clarity of your paper by explaining that "residents" refers to resident medical officers or doctors.

Line 75: Please review the references provided to support this claim (18 &19). In the preceding paragraph you have provided an overview of HL measurement tools, some of which have been developed since the references cited were published - a more contemporary critique of the issues associated with HL measurement is required. Linking this critique to the definitions of HL provided earlier in your paper would add to the cohesion and structure of your argument. 

Line 78: Consider rephrasing low HL as HL needs of patients. There is a diversity of HL needs within and between patients (people). As stated earlier in your paper, HL consists of a range of skills and abilities, so a person may have strengths in some areas and needs in another, therefore, to say the have "low HL" is reductionist and does not take account of this diversity.  

Line 79: Suggest delete "with low HL" - all patients have difficulty with jargon and technical terms (unless they are a HCP and even then, in the context of a stressful medical encounter, they may also find it difficult to understand). 

Line 81: Suggest that this be rephrased to identifying the HL needs of patients rather than low HL for reasons explained previously.

Line 85: Suggest that this be rephrased as identification of patients' HL needs (as explained previously). 

Line 87: As suggested in the feedback on the abstract of your paper, it is perhaps not reasonable to argue that HCP should be able to accurately estimate the HL of their patients. Suggest that this review and the previous research is generating evidence to the contrary, and rather than guess or estimate, HCP should be using appropriate HL measures to better understand the HL needs and preferences of their patients and ensuring that their practice and setting is HL responsive.

Line 90-93: Please re-word the purpose of your review for clarity. The explanation provided is very wordy and repetitive. 

Line 319: Please provide a reference for the EU study.

Line 331: Delete HL before "estimation"

Line 330-334: What evidence is there to suggest that other HCP working in other contexts (settings) would not also inaccurately estimate patient HL? And are these the studies that need to be recommended? See previous comments in relation to Line 87.

Line 335-336: There is a mismatch between "estimation" and "measurement". This occurs throughout your paper but this line is a clear example of this. You have referred to measurement tools, but then said they provide an estimate - is it an estimate if it is measured by the tool? The included studies refer to the HCP "estimating" the patients' HL as they did not measure it themselves or were asked to estimate without the benefit of knowing the measurement results. Please check this. 

Line 337-340: The link between the aspects of HL measured by the tools and the definitions of HL referred to in your paper is not very clear. See comments in relation to Line 75 as these are also relevant here. Please use the names of the three levels of health literacy for clarity (not first, second, third). It would also be useful here to critique the use of literacy and numeracy skills as proxy measures of HL. 

Line 344: Please check the clarity of this sentence - do you mean patients are more likely to communicate verbally than in writing during HCP encounters? 

Line 346: What do you mean by a lay theory of HL? You need to more clearly argue the point that you are trying to make about the mismatch between the HL measurement tools that focus on literacy and numeracy skills, and questions directly about literacy, and the HCP understanding of HL. This is an important point. Strengthening the earlier section in your paper with a more robust critique of HL measurement tools and the issues of measurement will in turn strengthen this argument.  

Line 348: What constructs? Although this question may be answered by reworking this paragraph as per the suggestions for Line 346.

Line 353: What do you mean by differently prone?

Line 351: Your recommendation for the use of the HLQ will be strengthened by following the suggestions earlier about the critique of other HL measurement tools. In addition to the explanation given, your recommendation also needs to explain the way the HLQ overcomes some of the limitations of other HL measures that rely on literacy and numeracy testing. And that the scales provide an indication of the needs, strengths and preferences ie an understanding of HL diversity.  In relation to this recommendation, you may also like to refer to a tool related to the HLQ, the CHAT (Conversational Health Literacy Assessment Tool) see: O’Hara, J., Hawkins, M., Batterham, R. et al. Conceptualisation and development of the Conversational Health Literacy Assessment Tool (CHAT). BMC Health Serv Res 18, 199 (2018). https://doi.org/10.1186/s12913-018-3037-6

Line 388: As per previous feedback, please reconsider the recommendation for HCPs to continue to "estimate" the HL of their patients. Rather, they need to better respond to the functional, interactive and critical health literacy needs of their patients; to be able to identify the health literacy needs, strengths and preferences; and to check understanding. 

References: Please check for consistency in journal titles. Spelling error in reference 17. 

Author Response

Response to Reviewer 2 Comments

Thank you for your review of the literature on this important topic. While previous research has already established that HCP overestimate the HL abilities of their patients, it is useful to have a current review of published studies. 

Abstract 

  1. Line 13-14: Given the findings of your review and previous studies of the accuracy (or lack thereof) of which HCPs estimate the health literacy of patients, it is perhaps not reasonable to argue that HCPS should be able to adequately estimate (guess) the health literacy needs of their patients. It is more reasonable to argue that HCPs should ensure they adequately communicate health information and check patient understanding. 

Thank you for your suggestion. We understand your point, but in our opinion assessing and identifying patient´ HL needs, checking patient understanding and the adequate communication are individual steps which all promote HCP-patient communication. The adequate assessment of patients HL needs by HCPs helps ensuring patient-centered health communication. When patients´ HL needs are adequately assessed, the communication of health information can be adapted. Constant checking by the HCPs to ensure that the information is understood are further steps in the process. Taking into account your comment, we changed the sentence in the abstract as follows:

“HCPs should rate patients’ health literacy (HL) needs and check patient understanding to ensure communicate adequate health information.”

We also discussed this point in the discussion as follows:

“These trainings should not only improve the identification of patients´ HL needs. There are further strategies to promote communication between HCPs and patients. These include the promotion of communications skills in general but also specific techniques, such as teach back. By the teach back method HCPs check patients understanding by asking them to state in their own words what has been explained to them by the HCP. Using this method, the patients understanding can be confirmed.”

  1. Key words: Suggest that you review these to increase the likelihood of your article being found by those interested in this topic. Consider those that you used to search for articles to include in your review. Avoid overlap with words in the title of your article. 

We revised the selection of keywords and decided on the following: HL; HCP; estimation; HL needs; physicians; patient-provider relationship; measurement

  1. Line 39-40: Check the explanation of functional health literacy. It is not limited to the ability to read and write - it also includes the ability to follow verbal instruction. 

Thank you very much for your comment. You are certainly right. Therefore we revised the explanation of functional HL: “The functional level includes basic reading and writing skills and being able to apply to health related materials or verbal health instructions”

  1. Line 44: Check the final sentence of the paragraph to ensure that it links to critical health literacy.

The description were adapted, the final sentence of the paragraph is now linked to the critical health literacy. “At the third level, the critical level, health information and existing recommendations are no longer arbitrarily accepted but increasingly questioned. At this level patients are already actively exchanging information with the existing healthcare system and are thus abandoning their passive role.”

  1. Line 46: HL is central to (delete for)

Thank you for your suggestion, we corrected the sentence accordingly. “HL is central to successful disease management and a prerequisite for the promotion of health-preserving and disease-preventing behaviour, for adequate handling of illnesses and for participation in the process of decision-making”

  1. Line 53-54: Consistent with contemporary understandings of health literacy, it would be more appropriate to refer to the purpose of measurement tools to identify the health literacy needs, strengths and preferences of patients rather than identify patients with low health literacy. 

Thank you for your suggestion, we corrected the sentence accordingly: “Measurement instruments assessing and identifying the health literacy skills, needs, strengths and preferences of patients are needed. “

  1. Line 69-73: Please provide a reference for the RQ - the reference provided (17) does not mention the RQ. It may add to the clarity of your paper by explaining that "residents" refers to resident medical officers or doctors.

This was expressed misleadingly. In the study by Zawilinski et al. (2019) the four resident questions are described as resident questionnaire (RQ). After further searching the literature, we have decided to describe the questionnaire as “resident questions”. The resident questions were developed as part of the Bass et al. (2002) study. In order to avoid misunderstandings, we have changed the description as follows:

“The resident questions that were utilized in the study of Bass et al. (2002) were designed to assess residents’ perception of patient HL levels. In this context the term 'residents' refers to resident medical officers or physicians. These can rate whether their patients have a HL problem, what level the patients´ HL is and whether and how the patients´ HL will impact the visits.”

  1. Line 75: Please review the references provided to support this claim (18 &19). In the preceding paragraph you have provided an overview of HL measurement tools, some of which have been developed since the references cited were published - a more contemporary critique of the issues associated with HL measurement is required. Linking this critique to the definitions of HL provided earlier in your paper would add to the cohesion and structure of your argument. 

With regard to your comment, we added the following text:

“In their review, Altin et al. (2014) examine recently published (2009 forward) papers dealing with the development and evaluation of HL instruments. Newly developed instruments apply multidimensional measurement of HL, whereas instruments developed before 2009 focus on basic skills regarding functional HL. Functional HL measurements overlooked the complexity of a patients´ HL and did not consider contextual factors, personal values, social resources and individual motivations that influence a patients´ ability to understand and act upon health information.”

  1. Line 78: Consider rephrasing low HL as HL needs of patients. There is a diversity of HL needs within and between patients (people). As stated earlier in your paper, HL consists of a range of skills and abilities, so a person may have strengths in some areas and needs in another, therefore, to say the have "low HL" is reductionist and does not take account of this diversity.  

Thank you for your comment. You are certainly right. Patients HL consist of a range of skills and abilities. Nonetheless, the instruments used in the included studies are mostly one-dimensional and cannot map the different domains of HL. We therefore considered the use of "HL needs of patients" instead of "low HL" in our review when it was appropriate for us. We also discuss the measurement with respect to low HL and HL needs:

“Over many years definitions for HL have evolved. Measurement of HL has proved complex because HL consist of different domains, skills and abilities. The instruments used in the included studies were mostly one-dimensional and did not reflect the different domains of patient HL. The aim of using the instruments in the included studies was to assign a high or low HL to patients. To measure independent domains of HL, we need multidimensional instruments. A multidimensional instrument is the Health Literacy Questionnaire (HLQ). The HLQ comprises nine scales that each measured a domain of the multidimensional construct of HL. Profiles can provide information about HL needs and strength. HCPs can use these profiles to better communicate and support the planning and realization of interventions. In a clinical context, HCPs can also use the Conversational Health Literacy Assessment Tool (CHAT) to identify patients’ multidimensional HL needs or preferences. Based on the domains of the HLQ, the CHAT was designed to support HCPs to use ten open-ended questions (e.g., Who do you usually see to help you look after your health?) to have a structured conversation with patients that target five HL areas. “

  1. Line 79: Suggest delete "with low HL" - all patients have difficulty with jargon and technical terms (unless they are a HCP and even then, in the context of a stressful medical encounter, they may also find it difficult to understand). 

We followed your suggestion and revised the sentence accordingly: “Patients have difficulties understanding technical terms and jargon, which are often embedded in medical communication during medical encounters.”

  1. Line 81: Suggest that this be rephrased to identifying the HL needs of patients rather than low HL for reasons explained previously.

Here we changed the sentence as follows:

“Misunderstandings could arise when HCPs have difficulties identifying the HL needs of patients.” Additional information is given in the answer to the comment 9 (line 78), too.

  1. Line 85: Suggest that this be rephrased as identification of patients' HL needs (as explained previously). 

Here we changed the sentence as follows:

“Training for HCPs could broaden the understanding of HL and thus support the identification of patients´ HL needs by HCPs.”

  1. Line 87: As suggested in the feedback on the abstract of your paper, it is perhaps not reasonable to argue that HCP should be able to accurately estimate the HL of their patients. Suggest that this review and the previous research is generating evidence to the contrary, and rather than guess or estimate, HCP should be using appropriate HL measures to better understand the HL needs and preferences of their patients and ensuring that their practice and setting is HL responsive.

Thank you for your comment. Here we changed the sentences as follows:

“Being able to correctly assess the patient’s HL and to identify patient’s HL needs is a prerequisite for patient-centred HCP-patient communication. Strategies including improved communication skills and using teach back methods further promote successful HCP-patient communication.”

  1. Line 90-93: Please re-word the purpose of your review for clarity. The explanation provided is very wordy and repetitive. 

Thank you for your comment. We re-word the purpose of our review for clarity:

“The purpose of this systematic review was to gain an overview of HL assessment in patients and HCPs. Specifically, the review aimed to describe (1) the assessment by patients and HCPs and (2) the agreement between patient and HCP HL assessment.”

  1. Line 319: Please provide a reference for the EU study.

Thank you for the suggestion. We added the following two references:

Kaper, M.S., Reijneveld, S.A., van Es, F.D., de Zeeuw, J., Almansa, J., Koot, J.A.R.,de Winter, A.F. Effectiveness of a Comprehensive Health Literacy Consultation Skills Training for Undergraduate Medical Students: A Randomized Controlled Trial. Int J Environ Res Public Health, 2019. 17.

Kaper, M.S., Winter, A.F., Bevilacqua, R., Giammarchi, C., McCusker, A., Sixsmith, J., Koot, J.A.R.,Reijneveld, S.A. Positive Outcomes of a Comprehensive Health Literacy Communication Training for Health Professionals in Three European Countries: A Multi-centre Pre-post Intervention Study. Int J Environ Res Public Health, 2019. 16.

  1. Line 331: Delete HL before "estimation"

Thank you for the comment, we revised the sentence accordingly.

“The studies included in this review mostly explored the agreement between physicians’ and patients’ estimation of patient HL.”

  1. Line 330-334: What evidence is there to suggest that other HCP working in other contexts (settings) would not also inaccurately estimate patient HL? And are these the studies that need to be recommended? See previous comments in relation to Line 87.

For a better understanding we revised the sentence as follows:

“Future studies should investigate other HCP groups, such as nurses or physiotherapists, to compare the identifying process of these HCP groups. Due to different educational background and access to patients, assessment of patients´ HL needs in these HCP groups may vary. Moreover, hospital and primary care settings were focused. There were no studies available that took place in other settings such as rehabilitation where HCPs spend more time with the patient.”

  1. Line 335-336: There is a mismatch between "estimation" and "measurement". This occurs throughout your paper but this line is a clear example of this. You have referred to measurement tools, but then said they provide an estimate - is it an estimate if it is measured by the tool? The included studies refer to the HCP "estimating" the patients' HL as they did not measure it themselves or were asked to estimate without the benefit of knowing the measurement results. Please check this. 

Thank you for the important suggestion; there was a linguistic mistranslation of the word "estimate/ estimation". We clarified the terms and decided to no longer use "estimate" or "estimation" in the context of measurement or standardized instruments throughout the manuscript. For example see line 3, 13, 64.

  1. Line 337-340: The link between the aspects of HL measured by the tools and the definitions of HL referred to in your paper is not very clear. See comments in relation to Line 75 as these are also relevant here. Please use the names of the three levels of health literacy for clarity (not first, second, third). It would also be useful here to critique the use of literacy and numeracy skills as proxy measures of HL. 

Thank you for your suggestion. We revised the description and used the names of the three level of HL instead of first, second, third level. Additional information about the link between the aspects of HL measured by the tools and the definition of HL is given in the answer to the comment 8 (line 75).

  1. Line 344: Please check the clarity of this sentence - do you mean patients are more likely to communicate verbally than in writing during HCP encounters? 

The wording of that sentence is misleading. For a better understanding we revised the description in the manuscript.

“The different levels of HL may also be differently accessible in the medical encounter: The functional HL level skills (e.g., reading abilities sufficient to understand and realize health information) may not be aspects that HCPs can estimate based on the medical encounter. The interaction is rather verbally than in writing. Patients` HL needs may not be detected by HCPs because the instruments used assess reading skills, but not aspects of verbal communication. Furthermore, HCPs may have their own subjective theories of what HL is. These could focus on aspects other than reading skills. When the HCP assessment of patient HL is determined by asking questions such as ‘‘Do you feel this patient has a literacy problem?’’, HCPs may rate based on their own subjective theories including specific domains or a multidimensional construct of HL. These are not covered by the HL instruments (i.e., REALM-R) which were completed by patients.”

  1. Line 346: What do you mean by a lay theory of HL? You need to more clearly argue the point that you are trying to make about the mismatch between the HL measurement tools that focus on literacy and numeracy skills, and questions directly about literacy, and the HCP understanding of HL. This is an important point. Strengthening the earlier section in your paper with a more robust critique of HL measurement tools and the issues of measurement will in turn strengthen this argument.  

Thank you for the important suggestion. Additional information is given in the answer to the comment 9 (line 78) and 20 (line 344), too.

  1. Line 348: What constructs? Although this question may be answered by reworking this paragraph as per the suggestions for Line 346.

The wording of that sentence is misleading. For a better understanding we revised the description: “When the HCP assessment of patient HL is determined by asking questions such as ‘‘Do you feel this patient has a literacy problem?’’, HCPs may rate based on their own subjective theories including specific domains or multidimensional constructs of HL. These are not covered by the HL instruments (i.e., REALM-R) which were completed by patients.”

Please see also the answer to the comment 20 (line 344).

  1. Line 353: What do you mean by differently prone?

We revised this section as follows:

“Future studies may use more comprehensive HL measures, such as the Health Literacy Questionnaire (HLQ). The agreement between the assessment of patients' HL by patients and HCPs may vary between the different scales. It should be tested whether the agreement between the assessment of patients' HL by patients and HCPs is changed or even improved by the use of multidimensional instruments.”

  1. Line 351: Your recommendation for the use of the HLQ will be strengthened by following the suggestions earlier about the critique of other HL measurement tools. In addition to the explanation given, your recommendation also needs to explain the way the HLQ overcomes some of the limitations of other HL measures that rely on literacy and numeracy testing. And that the scales provide an indication of the needs, strengths and preferences ie an understanding of HL diversity.  In relation to this recommendation, you may also like to refer to a tool related to the HLQ, the CHAT (Conversational Health Literacy Assessment Tool) see: O’Hara, J., Hawkins, M., Batterham, R. et al.Conceptualisation and development of the Conversational Health Literacy Assessment Tool (CHAT). BMC Health Serv Res 18, 199 (2018). https://doi.org/10.1186/s12913-018-3037-6

Thank you for the suggestion. We revised the description in the manuscript as follows: “Over many years definitions for HL have evolved [3]. Measurement of HL has proved complex because HL consist of different domains, skills and abilities. The instruments used in the included studies were mostly one-dimensional and did not reflect the different domains of patient HL. The aim of using the instruments in the included studies was to assign a high or low HL to patients. To measure independent domains of HL, we need multidimensional instruments. A multidimensional instrument is the Health Literacy Questionnaire (HLQ). The HLQ comprises nine scales that each measured a domain of the multidimensional construct of HL. Profiles can provide information about HL needs and strength. HCPs can use these profiles to better communicate and support the planning and realization of interventions. In a clinical context, HCPs can also use the Conversational Health Literacy Assessment Tool (CHAT) to identify patients’ multidimensional HL needs or preferences. Based on the domains of the HLQ, the CHAT was designed to support HCPs to use ten open-ended questions (e.g., Who do you usually see to help you look after your health?) to have a structured conversation with patients that target five HL areas.” (Please see also on comment 9, line 78)

  1. Line 388: As per previous feedback, please reconsider the recommendation for HCPs to continue to "estimate" the HL of their patients. Rather, they need to better respond to the functional, interactive and critical health literacy needs of their patients; to be able to identify the health literacy needs, strengths and preferences; and to check understanding. 

Thank you for the suggestion. Please note the answers to the previous comments 8 (line 75), 9 (line 78), and 24 (line 351) where the same topic was discussed.

  1. References: Please check for consistency in journal titles. Spelling error in reference 17. 

Thanks for the suggestion, we corrected the mistake.

Reviewer 3 Report

Overall this work is important. However, the presentation requires major revisions related to the comments below

Mechanics

The writing needs extensive editing. Too numerous to document. Many issues with phrasing, word choice and typographical errors. A line by line correction would take all day.

Content

Introduction

This paper lacks a theoretical basis. What is the theoretical relationship between patients self-report of healthcare literacy and the healthcare provider’s perception of the patients’ healthcare literacy? Are these generally congruent or not? Are they supposed to be congruent? What does the theory say about the connection between self-perception of literacy and actual literacy? There needs to be a discussion about an established relationship between the perception of literacy by the patient and the self-perception of literacy by the healthcare provider or lack their of and of discussion of the relationship between actual literacy and perception of literacy.

The literature review regarding the above is also absent. What current research has tested these relationships? What tools have bene used in the past to measure these concepts of perception of health literacy, actual literacy and have they been valid and reliable?

Overall the background does not describe the significance of the study thoroughly.

Methods

Study selection: The exclusion criteria is written awkward. Please reward. Eg. Studies were excluded if participants were less than 18 years of age, if a qualitative approach was used to measure literacy etc Same problem with inclusion criteria. The format for reporting these is sloppy

Measurement: Line 54. Remove the word “measuring” in front of instrument. Its redundant. Instrument implied measuring.

What is the validity and reliability of the instruments discussed???

Line 72. Should read “a” health literacy problem, not “an” health literacy problem.

Results: The results section needs to be organized with subheadings. And the results is just a repeat of what is in the table.

The results should address a research question posed in the beginning and as related to the theory?

Lines 287-297: You discussed the relationship between overestimation and underestimation and demographic variables. However, this analysis was not part of your introduction. Where is the theoretical discussion to support this or discussion of it significance? This should be in the introduction and therefore it would make sense for you to present this with the results. It just pops up unexplained.

Author Response

Response to Reviewer 3 Comments

Overall this work is important. However, the presentation requires major revisions related to the comments below

Mechanics

The writing needs extensive editing. Too numerous to document. Many issues with phrasing, word choice and typographical errors. A line by line correction would take all day.

Dear reviewer, thank you for the comments. Before we submitted the manuscript, it was professionally edited for proper English language, grammar, punctuation, spelling, and overall style (see certificate). This is the reason why we refrain from renewed English editing, but intensively checked new passages. .

You also criticized the choice of words was often inappropriate. There was a linguistic mistranslation of the word "estimate/estimation" in our manuscript. We clarified the terms and decided to no longer use "estimate" or "estimation" in the context of measurement or standardized instruments throughout the manuscript.

Content

Introduction

This paper lacks a theoretical basis. What is the theoretical relationship between patients self-report of healthcare literacy and the healthcare provider’s perception of the patients’ healthcare literacy? Are these generally congruent or not? Are they supposed to be congruent? What does the theory say about the connection between self-perception of literacy and actual literacy? There needs to be a discussion about an established relationship between the perception of literacy by the patient and the self-perception of literacy by the healthcare provider or lack their of and of discussion of the relationship between actual literacy and perception of literacy.

The literature review regarding the above is also absent. What current research has tested these relationships? What tools have bene used in the past to measure these concepts of perception of health literacy, actual literacy and have they been valid and reliable?

Overall the background does not describe the significance of the study thoroughly.

Thank you for your comment. In the following we would like to pick up your suggestions. The connection between the perception of HL by the patient and the perception of HL by HCPs was focused in the included studies and provides results on this topic. We cannot apply a theory. In the included articles the HCPS overestimate the patient HL. This also depends on the HL instruments used. The definition of objective and subjective assessment is described in more detail in the revision (line 63-91). We also argue that there are differences between self- and external/proxy perception, which is transferable to the assessment of patients' HL needs by patients and HCPs. The extent of agreement may give information with regard to patient-HCP-communication. This information may be even more valuable when multidimensional HL instruments are used. We revised the description of the introduction and discussion in order to clarify the relationship between the measurement instruments used and patient-HCP-communication, the differences between patients and HCPs HL assessment and the identification of patients HL needs through the use of multidimensional HL instruments. We also revised the description on validity and reliability of instruments used in the methods section.

Methods

Study selection: The exclusion criteria is written awkward. Please reward. Eg. Studies were excluded if participants were less than 18 years of age, if a qualitative approach was used to measure literacy etc Same problem with inclusion criteria. The format for reporting these is sloppy

We picked up your suggestion in the description of exclusion and inclusion criteria: “At title and abstract screening (screening phase), studies were excluded according to the criteria listed in Table 1. At full-text screening (eligibility stage) further inclusion criteria were added (see Table 1).

Measurement: Line 54. Remove the word “measuring” in front of instrument. Its redundant. Instrument implied measuring.

Thank you for the suggestion, we corrected the sentence.

What is the validity and reliability of the instruments discussed???

We picked up your suggestion and added the description of validity and reliability of the instruments in the methods section.

Line 72. Should read “a” health literacy problem, not “an” health literacy problem.

Thank you for the suggestion, we corrected the sentence.

Results: The results section needs to be organized with subheadings. And the results is just a repeat of what is in the table.

Thank you for your suggestions. The results are divided into five subthemes oriented to the purpose of the study. The text highlights specific results which are depicted in detail in the tables. The written results are difficult to understand without visualization.

The results should address a research question posed in the beginning and as related to the theory?

We re-word the purpose of our review for clarity:

“The purpose of this systematic review was to gain an overview of HCPs ability to assess HL levels in patients. Specifically, the review aimed to describe (1) the assessment by patients and HCPs and (2) the agreement between patient and HCP HL assessment.”

Lines 287-297: You discussed the relationship between overestimation and underestimation and demographic variables. However, this analysis was not part of your introduction. Where is the theoretical discussion to support this or discussion of it significance? This should be in the introduction and therefore it would make sense for you to present this with the results. It just pops up unexplained.

Thank you for the suggestion. We revised the introduction as follows: “Risk groups for having HL needs are non-native speakers, the elderly and those with limited education. These factors could also influence HCPs assessment of patient HL. “

Round 2

Reviewer 2 Report

Thank you for your positive response to the suggestions made previously. I now better understand that some of the issues noted were the result of translation.

I note that the word "estimate" has been removed and in most places replaced with "rate", however, this is not always appropriate and in some instances, it would perhaps be better to state "assess". For example, line 13 (abstract), it would be better to use the word "assess" as no specific health literacy rating tool is referred to here. 

Line 20-21: This sentence needs to be improved for clarity: "The HCPs rated patients’ HL by answering questions reflecting the content of standardized tools." Perhaps this is an occasion where "assessed" may be a better term. And changing to "that reflected" may improve the clarity.

Line 22-23: This is unclear " were overestimated by their HCPs in terms of their HL". Please re-word to indicate that the HCPs overestimated the HL level of patients. 

Line 35: spelling error - through, not though

Line 40-41: Thank you for more correctly explaining functional health literacy. However, the second part of this sentence suggests that people can only understand health information if they are functionally literate. 

Line 50-52: Please check the stem/beginning of this sentence to make sure the list of outcomes associated with low HL is clear. "...low HL have been associated with higher healthcare costs and poorer health outcomes, including but not limited to higher rates of ...."

Line 73: space missing between words, although suggest that "The resident" is deleted and the sentence starts at Questions...

Line 97: What do you mean by "... aswell as the different qualifications of the HCPs"?

Line 99: please check formatting; change to "HL assessment by patients
and HCPs"

Line 258: please amend for clarity: "...the patients rated their own HL". Suggest "...how the patients....". Similarly for line 259.

Line 318-319: please amend for clarity: "who has HL needsbased on their self-report were overestimated" - fix the spacing and make it clear that it is the HL level that is overestimated

Line 334: Please make this clearer by stating what you are referring to, rather than stating "it" ..."Further, it could decrease the rates of ..." what could decrease the rates?

Line 375: Please reword for clarity: "The different levels of HL may also be differently accessible in the medical encounter" What do you mean by differently accessbile? What levels of HL are you referring to - high or low or Nutbeam's dimensions?

Line 378: please correct the English language expression:  "The
interaction is rather verbally than in writing." 

Line 380: what do you mean by subjective theory? Is it better to try to explain that it might be a different interpretation of health literacy than patients etc. This issue occurs again in line 383.

Line 386-390: thank you for trying to incorporate some of the suggestion. However, this section needs to be carefully reviewed for clarity.

Line 412: correct was - replace with were (data plural)

Line 422: please correct to avoid any misunderstanding ie that this is not the first publication on the topic, rather it is a systematic review of the topic

Line 426-427: Please be clear that the training needs to include communication skills and knowledge, including checking understanding, as well as ways to identify health literacy needs (ie increase the HCP capacity to identify and respond to HL needs). 

Please proof read and edit your work carefully. 

Author Response

Reviewer 2 (Round 2)

Thank you for your positive response to the suggestions made previously. I now better understand that some of the issues noted were the result of translation.

  1. I note that the word "estimate" has been removed and in most places replaced with "rate", however, this is not always appropriate and in some instances, it would perhaps be better to state "assess". For example, line 13 (abstract), it would be better to use the word "assess" as no specific health literacy rating tool is referred to here. 

Thank you for your suggestion. We checked the relevant passages and replaced “rate” with “assess” when appropriate.

  1. Line 20-21: This sentence needs to be improved for clarity: "The HCPs rated patients’ HL by answering questions reflecting the content of standardized tools." Perhaps this is an occasion where "assessed" may be a better term. And changing to "that reflected" may improve the clarity.

Thank you. We revised the sentence according to your suggestions.

  1. Line 22-23: This is unclear " were overestimated by their HCPs in terms of their HL". Please re-word to indicate that the HCPs overestimated the HL level of patients. 

For a better understanding we revised the sentence as follows:

“Six studies reported that a high proportion of patients assigned to have HL needs based on their self-report were overestimated by their HCPs in terms of the HL level.”

  1. Line 35: spelling error - through, not though

We corrected the mistake.

  1. Line 40-41: Thank you for more correctly explaining functional health literacy. However, the second part of this sentence suggests that people can only understand health information if they are functionally literate.

We revised the sentence according to your suggestions.

“The functional level includes basic reading and writing skills and being able to apply to health-related materials or verbal health instructions. A patient who reaches this level is able to absorb and understand simple health-relevant information.”

  1. Line 50-52: Please check the stem/beginning of this sentence to make sure the list of outcomes associated with low HL is clear. "...low HL have been associated with higher healthcare costs and poorer health outcomes, including but not limited to higher rates of ...."

Here we changed the sentence as follows:

“A growing number of studies link low patients’ HL with higher healthcare costs and poorer health outcomes, including but not limited to higher rates of hospitalization and poorer medication adherence, poorer physical and psychological health and poorer self-management of chronic diseases (e.g., hypertension, diabetes) [5-9].”

  1. Line 73: space missing between words, although suggest that "The resident" is deleted and the sentence starts at Questions...

Thank you for your comment, we corrected the sentence.

  1. Line 97: What do you mean by "... aswell as the different qualifications of the HCPs"?

Thank you for your suggestion. We deleted this sentence.

  1. Line 99: please check formatting; change to "HL assessment by patients
    and HCPs"

Thank you for your suggestion. We checked the formatting and revised the sentence accordingly.

“The purpose of this systematic review was to gain an overview of HL assessment used in studies describing the agreement patients’ and healthcare professionals’ assessment of patients’ HL

  1. Line 258: please amend for clarity: "...the patients rated their own HL". Suggest "...how the patients....". Similarly for line 259.

Here we changed the sentence as follows:

“Overestimation meant that the HCPs assessed the patients' HL to be better than how the patients rated their own HL. In the case of underestimation, the HCPs assessed the patients' HL to be lower than how the patients rated their own HL.”

  1. Line 318-319: please amend for clarity: "who has HL needsbased on their self-report were overestimated" - fix the spacing and make it clear that it is the HL level that is overestimated

Thank you for the suggestion, we amend the sentence for clarity as follows:

“These six studies reported that the level of HL was overestimated by HCPs in a high proportion of patients having HL needs based on their self-reports (22%-58% of patients).”

  1. Line 334: Please make this clearer by stating what you are referring to, rather than stating "it" ..."Further, it could decrease the rates of ..." what could decrease the rates?

Thank for your comment, we deleted the sentence.

  1. Line 375: Please reword for clarity: "The different levels of HL may also be differently accessible in the medical encounter" What do you mean by differently accessbile? What levels of HL are you referring to - high or low or Nutbeam's dimensions?

Thank you for your comment. We reworded the sentences for clarity as follows:

“The different levels of HL may also be differently assessible by HCPs in the medical encounter: The functional HL level skills (e.g., reading abilities sufficient to understand and realize health information) may not be aspects that HCPs can observe in and estimate based on the medical encounter.”

One-dimensional measuring instruments can affect the assessment of the patients HL by HCPs. For example, instruments assessing functional HL focus on reading and writing level. As HCPs usually communicate orally with their patients, reading and writing skills are not directly observable in the medical encounter. .

  1. Line 378: please correct the English language expression:  "The
    interaction is rather verbally than in writing." 

We corrected the sentence as follows:

“As patients communicate verbally, patients’ HL needs identified by instruments that focus on reading skills (i.e., REALM-R) but not aspects of verbal communication, may not be detected by HCPs.”

  1. Line 380: what do you mean by subjective theory? Is it better to try to explain that it might be a different interpretation of health literacy than patients This issue occurs again in line 383.

Thank you for the suggestions. Here we changed the sentences as follows:

“Furthermore, HCPs may have their own conception of HL than patients.”

“HCPs’ conceptions of HL could focus on specific domains of the multidimensional construct of HL. Which domains the HCPs’ assessment is based on, cannot be determined by asking questions such as ‘‘Does your patient have low health literacy?’’. Moreover, HCPs may assess HL based on a specific domain of HL (e.g., Feeling understood and supported by healthcare providers).

  1. Line 386-390: thank you for trying to incorporate some of the suggestion. However, this section needs to be carefully reviewed for clarity.

Thank you for your suggestion. We reviewed this section and added the following sentences:

“Furthermore, HCPs may have their own conceptions of HL. HCPs’ conceptions of HL could focus on specific domains of the multidimensional construct of HL. Which domains the HCPs’ assessment is based on, cannot be determined by asking questions such as ‘‘Does your patient have low health literacy?’’. Moreover, HCPs may assess HL based on a specific domain of HL (e.g., Feeling understood and supported by healthcare providers). This may not be covered by the HL instruments (i.e., REALM-R) which were completed by patients. The multidimensional construct of HL should also be reflected when HCPs identify HL needs and strengths of their patients. Needs can be identified in one scale, there might be HL resources in another scale.”

  1. Line 412: correct was - replace with were (data plural)

We corrected the sentence.

  1. Line 422: please correct to avoid any misunderstanding ie that this is not the first publication on the topic, rather it is a systematic review of the topic

Thank you for your comment. For a better understanding we revised the sentence as follows:

“This systematic review summarizes the available literature and provides an overview on the topic of agreement between patients’ HL assess by patients and HCPs.”

  1. Line 426-427: Please be clear that the training needs to include communication skills and knowledge, including checking understanding, as well as ways to identify health literacy needs (ie increase the HCP capacity to identify and respond to HL needs). 

With regard to your comment, we added the following sentence:

“Furthermore, trainings need to impart knowledge on communication skills including checking understanding, as well as ways to adequately identify HL needs to increase the HCPs’ capacity to identify and respond to patients’ HL needs.”

  1. Please proof read and edit your work carefully. 

Thanks for your detailed and helpful suggestions. Finally, we checked our review intensively for the proper English language, grammar, punctuation, spelling and overall style.

Reviewer 3 Report

There continue to be minor grammatical errors throughout the paper too numerous to discuss them all. The are minor eliminations of the definite articles "a" and "the" and subject verb disagreements and paragraph structure and problems with organizing the sections of a research report. The content is very valuable and the technical part of the research of high merit. However, the writing and organization needs careful detailed attention. Would take me all day to edit and organize it. It needs line by line evaluation of mechanics and content to make sure sentences and paragraphs are in the right section and properly structures. It is essential information that needs to be published if the presentation can be sharpened.

Abstract.

Lines 11, 12, 13: The first sentence uses elementary language and does not use advanced language. The language should be elevated to a higher educational.  The first sentence can be changed to say.

“To navigate the healthcare system effectively, patients must be able to locate, comprehend and evaluate health information.”

The first two sentences in the abstract can be combined as such or something like this. They sound redundant since “find their way through” and “navigate” mean the same thing.

“Healthcare professionals (HCPs) can play a key role in promoting health literacy in patients to help them navigate the healthcare system effectively. This involves assisting them to locate, comprehend and evaluate health information.

Here, the definition of health literacy is essentially given.

Line 13: This sentence is grammaticall incorrect. You need the definite article “the” and patient needs an apostrophe “s” to show understanding.

HCPs should rate patients’ health literacy (HL) 14 needs and check patient understanding to ensure communicate adequate health information.

It should read this way:

“HCPs should rate patients’ health literacy (HL) 14 needs and check “the” patient’s understanding to ensure adequate communication of health information.”

Line 15: Needs the definite article “the” in front of patients.

Line 55: This discussion of measurements should be moved to the methods section. The discussion of healthcare provider assessment of literacy that begins in line 71 should follow directly after line 54’s discussion of the patients’ health literacy then make the connection between the patients health literacy and the provider’s assessment of their health literacy.

Iine 77-83’s discussion of instruments should go in the methods section when you talk about instruments. Take it out of the theoretical discussion.

The discussion in lines 85 to 101 where the connection between healthcare provider assessment and patient health literacy is made should come directly after the discussion of provider’s assessment of health literacy which is line 76.

Line 101. Make Sure Methods and Materils is a centered heading

The discussion of instruments should be under the methods section in a subheading titled instruments flushed to the left.

Author Response

Reviewer 3 (Round 2)

There continue to be minor grammatical errors throughout the paper too numerous to discuss them all. The are minor eliminations of the definite articles "a" and "the" and subject verb disagreements and paragraph structure and problems with organizing the sections of a research report. The content is very valuable and the technical part of the research of high merit. However, the writing and organization needs careful detailed attention. Would take me all day to edit and organize it. It needs line by line evaluation of mechanics and content to make sure sentences and paragraphs are in the right section and properly structures. It is essential information that needs to be published if the presentation can be sharpened.

Abstract.

  1. Lines 11, 12, 13: The first sentence uses elementary language and does not use advanced language. The language should be elevated to a higher educational.  The first sentence can be changed to say.

“To navigate the healthcare system effectively, patients must be able to locate, comprehend and evaluate health information.”

The first two sentences in the abstract can be combined as such or something like this. They sound redundant since “find their way through” and “navigate” mean the same thing.

“Healthcare professionals (HCPs) can play a key role in promoting health literacy in patients to help them navigate the healthcare system effectively. This involves assisting them to locate, comprehend and evaluate health information.

Here, the definition of health literacy is essentially given.

Thank you for your suggestion. With regard to your comment, we revised the first sentences according to your suggestion:

“Healthcare professionals (HCPs) can play a key role in promoting health literacy (HL) in patients to help them navigate the healthcare system effectively. This involves assisting patients to locate, comprehend and evaluate health information.”

  1. Line 13: This sentence is grammaticall incorrect. You need the definite article “the” and patient needs an apostrophe “s” to show understanding.

HCPs should rate patients’ health literacy (HL) 14 needs and check patient understanding to ensure communicate adequate health information.

It should read this way:

“HCPs should rate patients’ health literacy (HL) 14 needs and check “the” patient’s understanding to ensure adequate communication of health information.”

Thank you for the comment. We corrected the sentence accordingly.

“HCPs should assess patients’ health literacy needs and check the patient´s understanding to communicate adequate health information.”

  1. Line 15: Needs the definite article “the” in front of patients.

Here we changed the sentence as follows:

“This review investigates the agreement between the patients’ and HCPs’ assessment of patients’ HL.”

  1. Line 55: This discussion of measurements should be moved to the methods section. The discussion of healthcare provider assessment of literacy that begins in line 71 should follow directly after line 54’s discussion of the patients’ health literacy then make the connection between the patients health literacy and the provider’s assessment of their health literacy.

Thank you for your suggestion, we restructured the manuscript. In order to combine several suggestions, we moved the discussion of the measurement to the introduction section and to the discussion section. We did not move this part to the methods section, as it is not about the method of the review. The discussion of HCP assessment of literacy and the discussion of the patients’ HL were connected.

  1. Iine 77-83’s discussion of instruments should go in the methods section when you talk about instruments. Take it out of the theoretical discussion.

Thank you for the suggestion. Please note the answers to the previous comments 4.

  1. The discussion in lines 85 to 101 where the connection between healthcare provider assessment and patient health literacy is made should come directly after the discussion of provider’s assessment of health literacy which is line 76.

The discussion in lines 85 to 101 is now at the suggested position in the manuscript.

  1. Line 101. Make Sure Methods and Materils is a centered heading

Thank you for your comment. We checked and revised the formatting.

 The discussion of instruments should be under the methods section in a subheading titled instruments flushed to the left.

Thank you for the suggestion. Please note the answers to the previous comments 4.

Finally, we checked our review intensively for the proper English language, grammar, punctuation, spelling and overall style.
